# Nutritional Biomarkers for the Prediction of Response to Anti-TNF-α Therapy in Crohn’s Disease: New Tools for New Approaches

**DOI:** 10.3390/nu16020280

**Published:** 2024-01-17

**Authors:** Fernando Rizzello, Ilaria Maria Saracino, Paolo Gionchetti, Maria Chiara Valerii, Chiara Ricci, Veronica Imbesi, Eleonora Filippone, Irene Bellocchio, Nikolas Konstantine Dussias, Thierry Dervieux, Enzo Spisni

**Affiliations:** 1IBD Unit, IRCCS, Azienda Ospedaliero-Universitaria di Bologna, University of Bologna, Via Dr. Massarenti 9, 40138 Bologna, Italy; fernando.rizzello@unibo.it (F.R.); paolo.gionchetti@unibo.it (P.G.); veronica.imbesi3@unibo.it (V.I.); eleonora.filippone2@unibo.it (E.F.); nikolas.dussias@studio.unibo.it (N.K.D.); 2Department of Medical and Surgical and Sciences, University of Bologna, Via Dr. Massarenti 9, 40138 Bologna, Italy; 3Microbiology Unit, IRCCS, Azienda Ospedaliero-Universitaria di Bologna, University of Bologna, Via Dr. Massarenti 9, 40138 Bologna, Italy; ilariamaria.saracino@studio.unibo.it; 4Department of Biological, Geological and Environmental Sciences, University of Bologna, Via Dr. Selmi 3, 40126 Bologna, Italy; irene.bellocchio2@studio.unibo.it (I.B.); enzo.spisni@unibo.it (E.S.); 5Gastroenterology Unit, ASST Spedali Civili di Brescia, University of Brescia, Piazza del Mercato 15, 25121 Brescia, Italy; chiara.ricci@unibs.it; 6Prometheus Laboratories, 9410 Carroll Park Dr., San Diego, CA 92121, USA; tdervieux@prometheuslabs.com

**Keywords:** Crohn’s disease, anti-TNF-α biological therapy, remission phase, active phase, vitamins, mineral, nutrient deficiency

## Abstract

Crohn’s disease (CD) is a chronic disorder of the digestive tract characterized by an uncontrolled immune-mediated inflammatory response in genetically predisposed individuals exposed to environmental risk factors. Although diet has been identified as one of the major environmental risk factors, the role of nutrients in the clinical management of CD patients has not yet been fully investigated. In this prospective observational study, fifty-four patients diagnosed with active Crohn’s disease and undergoing anti-TNF-α biological therapy were enrolled and subjected to nutrient intake analysis through a daily food diary. Their nutrient intake and blood values were analyzed before and after 6 months of biological therapy. After 6 months of anti-TNF-α, four patients dropped out of the study, leaving 29 patients in clinical remission and 21 still with active disease that remained the same. The aim of this study was to identify nutrients whose intake or blood values may be associated with patients’ responses to biological therapy. In the diet, patients remaining with active CD showed very similar nutrient dietary intake compared to patients achieving remission except for a trend for lower starting zinc intake, below the reference value. In the blood, instead, patients who did not respond to biological therapy showed significantly lower plasma values of iron and taurine before starting biological anti-TNF-α treatment.

## 1. Introduction

Inflammatory bowel diseases (IBD) comprise Crohn’s disease (CD) and ulcerative colitis (UC), two chronic, inflammatory disorders of the digestive tract that develop especially in adolescence and early adulthood and overall affect at least 1.5 million Americans and 2.2 million Europeans [1]. IBD are characterized by an uncontrolled immune-mediated inflammatory response in genetically predisposed individuals exposed to environmental factors, collectively referred to as the exposome. The exposome concept includes diet, medications, nicotine, infectious agents, stress, and lifestyle. These collectively contribute to triggering gut chronic inflammatory loops [2,3]. The shift of the population towards Western-style dietary habits, called the Western diet, is considered an important risk factor for IBD [4,5]. Nevertheless, the role of nutrients in the clinical management of CD patients has not yet been fully investigated. Data reported in the literature mainly concern surveys on dietary habits in patients with IBD or focus on interventional studies in CD patients aimed at evaluating the effectiveness of specific dietary protocols to aid or to maintain clinical remission [6]. Our research group had already highlighted substantial differences in the nutrient intake of subjects with Crohn’s disease in comparison with healthy subjects [7]. In particular, the cross-sectional analysis of the eating habits of these patients showed that CD patients significantly reduced the consumption of fiber and vitamins (A, E, C, B6, folic acid) with respect to healthy subjects. Furthermore, their calcium, potassium, phosphorus, iron, magnesium, copper, and iodine intake was significantly lower [7].

The lack of therapeutic response is a common problem for IBD patients during treatment with anti-TNF-α agents [8], which are the most commonly used drugs for the treatment of these diseases. Thus, treatment with these biologics must be optimized, and treatment strategies to increase the therapeutic response to anti-TNF-α drugs represent an important goal to improve the prognosis of IBD patients.

It has been reported that a concomitant elementary diet is capable of decreasing the loss of response to infliximab or adalimumab (which are the most used anti-TNF-α drugs) in patients with Crohn’s disease [9,10]; nevertheless, no controlled studies have been designed to analyze nutrient intake or nutrient blood values as possible factors linked to the lack of response to the anti-TNF-α biological therapy. The aim of this prospective study was to compare the nutrient intake and the nutritional status of patients who, after six months of anti-TNF-α biological therapy, were still responders with those of patients who remained with active CD. The primary endpoint of this study was to understand whether patients’ nutrient intake and nutrient blood levels could be related to their response to the anti-TNF-α biological therapy, and the secondary endpoint was to understand how the induction of the remission phase modified patients’ nutrient intake and nutrient blood levels. Until now, very few clinical studies have examined the association between nutrients and the clinical response to anti-TNF-α. The present study aimed to investigate this possible correlation.

## 2. Materials and Methods

### 2.1. Study Population

Fifty-four adult patients (European, male/female, 18–65 years old) with moderate–severe active Crohn’s disease and with an indication for anti-TNF-α therapy according to the normal clinical practice, and thirty no-IBD controls (European, male/female, 18–65 years old) with no gastrointestinal disorders, as defined by medical history and standard clinical biochemistry values, afferent to the outpatient clinic, started the clinical trial. CD patients anti TNF-alpha therapy was administered by infusion, thus compliance to the therapy was closely monitored by a medical specialist. The study was conducted at the Regional Reference Center for IBD of the IRCCS Policlinico Sant’Orsola-Malpighi, Bologna, Italy, and was approved by the Regional Ethics Committee (CER): Study code 16/2015/U/Tess approved on 10 February 2015. All patients signed an informed consent before starting any study procedure. The study was conducted in accordance with the principles of the Declaration of Helsinki and Good Clinical Practice.

### 2.2. Study Design

This study was performed in the context of a one-year prospective observational clinical trial aimed at identifying biomarkers and predictors of response to commonly used biological therapy in patients with Crohn’s disease. The full protocol with all study procedures and analysis is available at the site www.clinicaltrial.gov, ClinicalTrials.gov Identifier: NCT02580864, accessed on 19 October 2015.

### 2.3. Inclusion Criteria

To be eligible to participate in the study, participants had to have signed informed consent and be aged between 18 and 65. They needed to have been diagnosed with CD at least 3 months before screening (involving the small intestine and/or colon), which needed to be confirmed by standard criteria (clinical, endoscopy, laboratory, histology, and/or radiology). Participants needed to be of European ethnicity with indications for anti-TNF-α therapy.

### 2.4. Exclusion Criteria

Ineligibility criteria for participation in the study included the following: Pregnancy or breast-feeding (on the date of the visit), participation in any CD-related clinical trial at the time of enrolment or during the 3 months before the visit, enterocutaneous, abdominal, or pelvic active fistulae with abscesses or fistulae likely to require surgery during the study period, bowel surgery, other than appendectomy, within 12 weeks prior to randomization and/or surgery planned or deemed likely for CD patients during the study period. Other exclusion criteria include a history of extensive colonic resection and subtotal or total colectomy. The presence of ileostomies, colostomies, or rectal pouches, a history of more than three small bowel resections or diagnosis of short bowel syndrome, known clinically significant stenoses, history or evidence of adenomatous colonic polyps that have not been removed, history or evidence of colonic mucosal dysplasia, chronic use of narcotics for chronic pain, defined as daily use of one or more doses of narcotic-containing medication, prior use of biologic therapy including adalimumab and infliximab, antibiotic treatment within 2 weeks prior to baseline, and tube or enteral feeding, elemental diet, or parenteral nutrition within 4 weeks prior to baseline. 

### 2.5. Study Procedures

The study procedures are briefly summarized. During visit V1 (before starting biological therapy) and V3 (after six months of biological therapy), CD patients completed a food frequency questionnaire (Appendix A), which was filled out throughout the seven days before the scheduled visit. A nutritionist reviewed the questionnaires with the patient during each scheduled visit to ensure the accuracy of the information provided. No nutritional interventions were performed on patients. Moreover, at each visit, blood analysis on fasting patients and healthy subjects was performed to detect plasma cholesterol, vitamin B12, vitamin B9, minerals (K, Mg, Fe), and amino acids. The study design is reported in Figure 1.

### 2.6. Food Questionnaires and Nutrient Analysis

Records obtained by food diaries at V1 and V3 were entered into a specific nutritional software (Metadieta, METEDA srl, Rome, Italy, software version 4.6) to break down foods into nutrients (performing a bromatological analysis). This software quantifies macro- and micronutrient contents of food based on the Italian food composition database described by Gnagnarella and colleagues [11]. 

Data obtained were then exported to an Excel database. For each patient, a 7-day mean of each nutrient intake was calculated. Nutrients considered for bromatological analysis were proteins (total, animal, and vegetable), amino acids, saturated and unsaturated fatty acids, carbohydrates (available carbohydrates, amid, oligosaccharides), cholesterol, total fibers, soluble and insoluble fibers, minerals (calcium, iron, sodium, potassium, chloride, chromium, iodine, fluorine, magnesium, copper, selenium), vitamins (A, B1, B2, B6, B9, B12, C, D, E) and total polyphenols. Total daily calories were also calculated. Furthermore, through Metadieta software analysis, we were able to obtain data about the ORAC (oxygen radical absorbance capacity) and PRAL (potential renal acid load) of the diets. ORAC is an index that estimates antioxidant food activities [12]; PRAL is an indirect measure of the biochemical balance of acidifying and alkalizing molecules contained in foods [13].

### 2.7. Statistical Analysis

Continuous variables are expressed as mean ± standard deviation (SD) or median and interquartile range (IQR), accordingly. The unpaired Mann–Whitney U test was used for the comparison between groups for both nutrient intake and blood values. The paired Wilcoxon signed-rank test was used to compare patient-paired data between V1 and V3. Comparisons among patient subgroups were performed using the chi-squared test (Fisher’s exact test when appropriate). *p*-values of <0.05 were considered statistically significant, while *p*-values of <0.10 were considered as a statistical trend considering the size of the sample. Analyses were performed using IBM SPSS statistic version 28 (IBM SPSS Statistics, Armonk, NY, USA: IBM Corporation).

## 3. Results

### 3.1. Study Population

The demographic characteristics of CD patients are reported in Table 1. In the same table, patients have been stratified based on their therapeutic response after 6 months of anti-TNF-α therapy: responders (CD-R) and non-responders with active disease (CD-A). Of the 54 CD patients enrolled in the study, four CD patients withdrew from the trial after three months of follow-up (V2) due to problems related to the excessive distance of the clinical center from their residence. Thus, only data from 50 CD patients were analyzed. All enrolled patients had active CD at the first study visit V1, while at V3, after 6 months of biological therapy, 29/50 patients (58%, 95%CI 44.23–40.62) were in the remission phase defined by the Crohn’s Disease Activity Index (CDAI) < 150, while 21/50 (42%, 95%CI 29.4–55.7) still had active CD. Based on the CDAI score, patients were divided into two subgroups, that of responders (remission, CD-R) and that of non-responders (still active disease, CD-A) to the biological anti-TNF-α therapy. Analyses of CD patients at V1 have also been stratified, dividing future responders into future non-responders to analyze possible initial clinical differences between the group of patients who will respond to anti-TNF-α therapy (CD-R) and the group who will not achieve a therapeutic response (CD-A).

### 3.2. Macronutrient Intake

The total daily caloric intake (measured in Kcal/day) showed no significant differences in the comparison between CD-A (1799.76 ± 647.23 Kcal/day) and CD-R (19,710.84 ± 729.15 Kcal/day) at V3. The same groups analyzed at V1 showed similar caloric intake before the beginning of the biological therapy (CD-R at V1: 1936.31 ± 736.89; CD-A at V1: 1738.99 ± 529.58 Kcal/day). The distribution of dietary macronutrients (Figure 2) did not show significant differences between the two groups both at V3 and V1 and remains within the reference ranges of intake suggested for the Italian population by the dietary reference values published by the European Food Safety Authority (EFSA-DRVs).

The two groups did not even show differences in the amounts of proteins of animal and vegetable origin. Intake of proteins of animal origin was 36.68 g/day (IQR 31.6–47.07) in CD-A and 41.38 g/day (IQR 33.59–50.79) in CD-R (*p* = 0.49). Intake of protein of vegetal origin was 20.23 g/day (IQR 17.56–30.37) in CD-A and 23.58 g/day (IQR 15.07–28.12) in CD-R (*p* = 0.86). The ratio between animal proteins/vegetable proteins was found to be 1.7 (IQR = 1.22–2.02) in CD-A and 1.93 (IQR = 1.36–2.35) in CD-R, not statistically different (*p* = 0.44 for comparisons between ratios).

The percentage of total daily calories derived from ultra-processed (UP) foods that may interfere with IBD progression or remission [6] were not significantly different comparing patients in remission to those still active at V3 and at V1 and ranged between 15% and 17% at V1, and between 20% and 22% at V3 (Appendix A–D). These values are in line with the average value of 13% reported in 2022 for the Italian population [14]. Even when the analysis of macronutrients does not show significant differences in intake, the choices of food groups used to cover nutritional needs can be very different. For this reason, we decided to verify whether there were differences in the main food groups that made up the diets of the patients in our court. Analysis of food groups showed that at V3, patients with active disease consume lower amounts of dairy products. At V1, we observed only small differences with a significantly higher white meat intake in CD-A and, in the same group, a trend for lower fish consumption (Table 2 and Table 3).

### 3.3. Vitamins and Minerals Intake

In terms of vitamin intake, patients with CD in the active phase showed no differences if compared to those with CD in the remission phase, both at V3 and V1. For minerals, 15 different elements were analyzed: calcium, sodium, potassium, phosphorus, iron, zinc, magnesium, copper, selenium, chromium, fluorine, iodine, magnesium, molybdenum, and nickel. Significant differences were observed between the two patient groups only in calcium intake at V3, with CD-R showing a significantly higher intake with respect to the CD-A group. At V1 instead, no statistically significant differences were evident with only a trend for a decreased zinc intake in CD-A patients (Table 4, Table 5, Table 6 and Table 7).

### 3.4. Lipids

At V3, we observed a significantly increased intake of capric acid and a trend for an increased intake of lauric acid in patients who achieved remission. This increase should be related to an increase in the consumption of dairy products rich in medium-chain triglycerides (MCT). Analysis at V1 only showed a trend for an increased intake of myristoleic acid in CD-A patients. Lipid analyses are detailed in Table 8 and Table 9.

### 3.5. ORAC, PRAL and Polyphenols

The whole diet’s oxygen radical absorbance capacity (ORAC) was very similar in both the CD groups at V3 and V1 (see Appendix A). The potential renal acid load (PRAL) of the diet was also quite similar in both CD groups without significant differences between CD-R and CD-A at both time points, and the same happened for the total polyphenol intake (See Appendix A). It should be noted that all CD patients had a high PRAL level, which was always above 15 pr/day. This indicates an imbalanced diet with a high ratio of animal to vegetable proteins.

### 3.6. Blood Nutrient Values

The comparison between CD-A and CD-R at V3 showed significantly decreased plasma levels of vitamin B9, magnesium, and iron in patients with active disease (Table 10 and Table 11). Moreover, these patients also showed significantly decreased plasma levels of nine different amino acids (threonine, glutamic acid, citrulline, valine, cysteine, methionine, phenylalanine, histidine, and arginine) and a statistical trend for three others (leucine, tyrosine, lysine). Also, vitamin B12 showed a decreased trend in CD-A patients. These data find an easy explanation for the worse intestinal absorption of nutrients that always occurs in patients with active disease compared to those in clinical remission.

Blood analysis at V1, before the beginning of anti-TNF-α therapy, showed that there were significantly decreased levels of iron and taurine and a trend for a decreased vitamin B12 in the group of patients who would not achieve clinical remission. Analyzing in detail the iron and taurine values, which were significantly different between the CD-A and CD-R groups (Figure 3), we highlight that patients with iron and taurine values less than or equal to the 1st decile were all active at V3 (0% responders). On the contrary, all the patients with iron and taurine values greater than or equal to the 9th decile achieved remission at V3 (100% of responders). Analyzing this difference in therapeutic response between the first and the 9th iron–taurine decile by using Fisher’s exact test, the result is that the two groups are statistically different with a *p* = 0.002 (*p* < 0.01), which means a very statistically significance despite the low number of patients. The comparison of the same two groups of patients considering the severity of the disease at V1 shows no significant differences, nor with regard to the SES-CD (SES-CD 1st decile mean = 9.17, SD = 3.49, Mdn = 8, IQR = 6–13; SES of the 9th decile mean = 9.20, SD = 3.03, Mdn = 11, IQR = 7–11; *p* = 0.92), nor with regard to the CDAI (CDAI of the 1st decile mean = 194.60, SD = 85.27, Mdn = 231, IQR = 109–260; CDAI of the 9th decile mean = 134.25, SD = 97.6, Mdn = 130, IQR = 50–218.5; *p* = 0.19).

## 4. Discussion

The analysis of daily calorie and macronutrient intakes reveals no significant differences between the two groups of patients (CD-A and CD-R), both six months into anti-TNF-α therapy (V3) and prior to its initiation (V1). Also, caloric intake from ultra-processed foods, with several of their ingredients having negative effects on the intestinal microbiome [15], was completely similar between the two groups. However, the analysis of the nutritional patterns shows some differences, with a lower calcium intake in the CD-A group at V3. This data is explainable because dairy products are among the foods that are self-reduced by CD patients as they are considered responsible for part of the symptoms [16], probably the IBS-like symptoms, from which these patients frequently suffer. With the persistence of the symptoms, it is understandable that the CD-A patients decreased the consumption of these foods at V3. At V1, before starting anti-TNF-α therapy, patients who did not achieve remission showed a trend for a reduced intake of zinc, whose average in CD-A resulted in being lower than the reference value for adults (8 mg/day). For this nutrient, we have not analyzed the blood values, and therefore, we cannot confirm to what extent the reduced intake corresponded to reduced blood values. Nevertheless, zinc deficiency prevalence seems to be quite high among IBD patients, especially in CD [17,18]. The zinc level in the blood has already been related to a worse clinical progression in CD patients. Thus, it has been suggested that the adoption of zinc deficiency is a predictor of CD outcomes [19,20].

Lipid intake showed very small differences between the CD-A and CD-R groups. At V3, we observed only a reduced intake of medium-chain fatty acids in CD-A, which was significant for capric acid and a trend for lauric acid. This is in line with the reduced intake of calcium and dairy products rich in MCT. Analysis at V1 showed a trend for an increased intake of myristoleic acid in patients who would remain with active disease.

The most interesting data from this study comes from the nutrient blood analyses of these patients. In fact, at V3, the group of patients with the disease still in the active phase showed a series of significant decreases in plasma nutrient levels that included nine amino acids (threonine, glutamic acid, citrulline, valine, cysteine, methionine, phenylalanine, histidine, and arginine), vitamin B9, magnesium, and iron. Moreover, we observed a trend for a decrease in vitamin B12 and three other amino acids (leucine, tyrosine, and lysine). These decreases in circulating nutrients must be considered a direct consequence of the active disease. They are in line with results obtained in other studies defining the multiple nutrient deficiencies caused by the active CD that impair their intestinal absorption [21,22]. This so-called inflammation-related malnutrition has been linked to the alteration of the absorptive surface of the small intestine and also to the higher consumption of amino acids, minerals, and vitamins necessary to maintain an overactivated immune system [22,23,24]. Therefore, the decrease of these nutrients in the plasma of CD-active patients is not surprising, with levels that, in some cases, were below the reference values.

Analyzing patients’ blood nutrient levels before starting the biological therapy (V1), we found that those who did not achieve remission by using anti-TNF-α had a significantly lower level of iron and taurine and only a trend for decreased vitamin B12 levels. These differences are not attributable to more severe disease in this patient group; in fact, from a clinical point of view, there were no differences at V1 in disease severity between patients with iron and taurine values less than or equal to the first decile in comparison with a patient with iron and taurine values greater than or equal to the 9th decile. Moreover, there were no differences between CD-A and CD-R in iron or in animal protein intake at V1 that could explain different blood levels of iron and taurine. Even if these findings could be linked to the large number of variables analyzed, possible physiopathological reasons for these observed differences reside in the intestinal absorption efficiency of these nutrients or in gut microbiota composition (and metabolism) that is capable of affecting their absorption [25,26,27,28].

The role of taurine in IBD is quite controversial; in fact, on the one hand, taurine is considered anti-inflammatory and antioxidant [29,30,31], and its administration in animal models of IBD seems to have a protective effect [30,31]. On the other hand, in IBD patients, a higher concentration of taurine in fecal samples, together with a rise of taurine bacterial metabolism, has been related to a higher risk of more severe disease, probably because an end product of taurine bacterial metabolism is hydrogen sulfide which is able to damage enterocytes and colonocytes [27,32]. Thus, the differences between what has been observed in animal models and what has been observed in patients could be linked to the microbial metabolism of taurine that could happen in the profoundly altered microbiota of IBD patients. In this study, we have not collected information on taurine gut metabolism; therefore, we have no data on fecal taurine levels.

Iron deficiency has been studied for a long time in IBD, and it has been observed in patients with chronic bleeding [33]. Additionally, a strong immune system activation could lead to iron sequestration in the mononuclear cells [34]. Iron levels have already been linked indirectly to the therapeutic response to biological drugs [35]. Indeed, hepcidin, a good iron-deficiency biomarker, was found to be significantly decreased in IBD patients who responded to the anti-TNF-α therapy [36,37]. Our patients were homogeneous both from a clinical and endoscopic point of view, and we have no reason to hypothesize an increased immune activation in CD-A. This hypothesis should be confirmed by histological examination of their gut wall.

As regards the dietary changes induced by the remission of the disease, we have highlighted how patients who enter remission tend to increase the consumption of dairy products and, consequently, their intake of calcium and medium-chain fatty acids.

We are aware of the limitations of this study, which concern the indirect determination of nutrient intakes and the limited number of patients that do not allow for validation of a cut-off value of iron and taurine that can be considered as independent and predictive risk factors for the response to anti-TNF-α therapy. Nevertheless, our data suggest that iron and taurine blood levels may be somehow connected to the therapeutic outcome of CD patients to biological therapy.

These preliminary data constitute a good rationale for designing a larger study to eventually confirm the relationship between iron and taurine blood levels and the therapeutic response to anti-TNF-α drugs.

## 5. Conclusions

This study highlights that there may be circulating nutrients easily detectable by blood tests, such as iron and taurine, that could act as independent biomarkers capable of improving the prediction of the therapeutic response of CD patients to anti-TNF-α therapies. These biomarkers could be used in association with those already validated and in use to assist therapeutic decision-making. Larger studies will be needed to confirm these data and possibly validate these predictive nutrient biomarkers.

## Figures and Tables

**Figure 1 nutrients-16-00280-f001:**
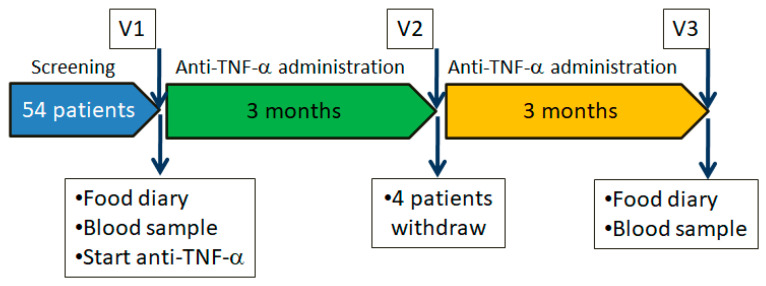
Study design.

**Figure 2 nutrients-16-00280-f002:**
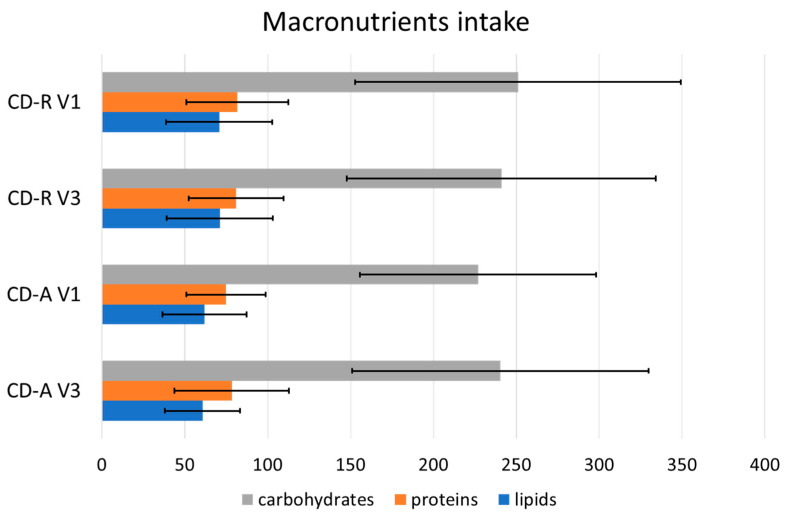
Macronutrient consumption indicated in grams per day (±standard deviation) in CD-R (clinical remission) and CD-A (active disease) groups at V3 and V1.

**Figure 3 nutrients-16-00280-f003:**
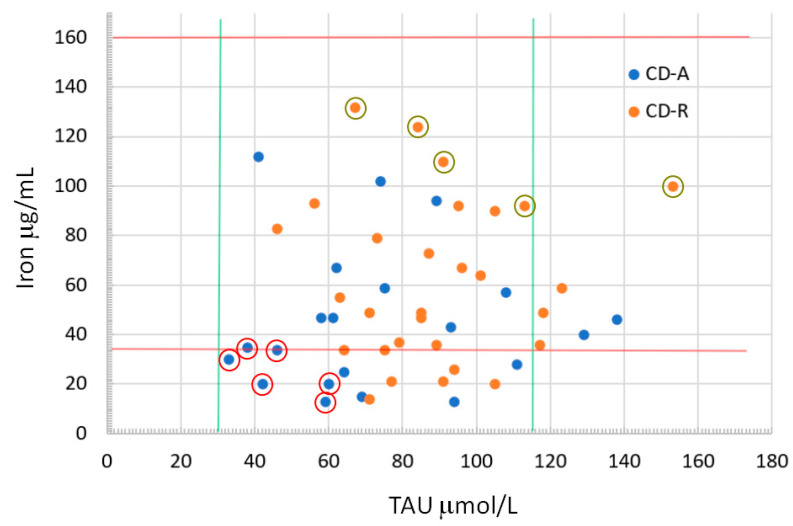
CD patients represented in a Cartesian plane with iron values on the ordinate and taurine values on the abscissa. The red and green lines represent the minimum and maximum values of the relative reference blood range (37–160 mg/mL for iron; 30–116 mmol/L for taurine). The two groups of CD-A and CD-R patients clearly do not form a cluster. The red circles indicate patients with iron and taurine values less than or equal to the first decile, while dark green circles indicate patients with iron and taurine values greater than or equal to the 9th decile.

**Table 1 nutrients-16-00280-t001:** Demographics in the study cohort of Crohn’s disease patients (CD) at V1 were divided for their therapeutic response to anti-TNF-α in CD-R and CD-A.

	CD (n:50) N (%) or Mdn (IQR)	CD-R (n:29) N (%) or Mdn (IQR)	CD-A (n:21) N (%) or Mdn (IQR)	P (CD-A vs. CD-R)
Gender				
Female	18 (36.0)	9 (31.1)	9 (42.8)	0.39
Male	32 (64.0)	20 (68.9)	12 (57.2)	
Age				
Female	39.1 (20–55)	43.0 (42.0–48.0)	33.0 (21.0–45.0)	0.19
Male	35.7 (20.0–64.0)	32.5 (24.0–40.5)	34.5 (27.0–45.0)	0.47
BMI				
Female	22.3 (18.0–25.0)	23.62 (22.48–24.91)	20.76 (19.59–23.53)	0.22
Male	22.8 (18.0–25.0)	23.84 (21.2–25.9)	23.71 (22.17–26.4)	0.58
Smokers	11	8	3	0.26
CDAI	150.0 (99.0–209.0)	134.0 (86.5–201.0)	172.0 (145.5–220.0)	0.12
SES CD	10.0 (7.0–14.0)	11.0 (7.0–14.0)	8.0 (7.0–13.0)	0.56
Symptoms duration *	8 (0.57–14.0)	8 (0.75–13.0)	6 (0.36–14.5)	0.92

* Expressed in months.

**Table 2 nutrients-16-00280-t002:** Dietary intake values (g/day) reported as median and interquartile range (IQR).

	CD-A V3	CD-R V3	CD-A V1	CD-R V1
Soluble fibers	2.26 (1.50–3.00)	2.40 (1.70–1.350)	2.45 (1.31–2.87)	2.25 (1.64–2.85)
Insoluble fibers	3.02 (2.27–5.63)	4.54 (2.42–6.90)	4.24 (2.71–5.22)	3.84 (2.99–6.03)
Milk	0.00 (0.00–85.75)	20.00 (0.00–108.95)	0.00 (0.00–50.71)	5.71 (0.00–46.43)
Dairy products	25.60 (5.70–55.35)	52.86 (27.50–78.57)	47.86 (19.29–77.14)	55.71 (34.29–96.07)
Fish	17.1 (0.00–42.10)	17.14 (11.43–52.86)	7.14 (0–12.86)	15.71 (0.00–42.86)
White meat	25.60 (5.7–55.35)	21.43 (7.86–38.57)	42.86 (16.43–62.14)	14.29 (0.00–32.14)
Red meat	51.40 (28.60–62.85)	68.57 (23.57–97.14)	52.86 (20.71–76.43)	68.57 (27.86–114.64)
Processed meat	38.57 (17.14–60.00)	25.00 (14.29–42.14)	50.00 (29.29–82.14)	37.14 (17.14–50.00)
Eggs	0.00 (0.00–4.30)	0.00 (0.00–15.71)	0.00 (0.00–14.29)	0.00 (0.00–17.14)
Starchy foods ^#^	232.90 (202.85–298.55)	245.29 (141.64–319.79)	262.86 (182.5–317.86)	246.14 (142.86–299.64)
Fruit	47.86 (4.29–195.00)	85.71 (28.57–155.71)	69.28 (28.57–121.43)	81.43 (21.43–112.86)
Vegetables	75.36 (40.00–132.86)	85.71 (44.29–135.71)	60.35 (25.71–102.86)	85.71 (40.00–165.71)

^#^ = containing gluten.

**Table 3 nutrients-16-00280-t003:** Statistical significance of differences in dietary intake in patients in remission (CD-R) and in active phase (CD-A) at V3 and V1. *p* Values (two-tailed) are reported.

	V3 CD-A vs. CD-R		V1 CD-A vs. CD-R	
Soluble fibers	0.54		0.99	
Insoluble fibers	0.44		0.65	
Milk	0.58		0.47	
Dairy products	0.04 *	CD-A < CD-R	0.39	
Fish	0.16		0.05 **	CD-A < CD-R
White meat	0.93		0.03 *	CD-A > CD-R
Red Meat	0.65		0.21	
Processed meat	0.22		0.16	
Eggs	0.10		0.85	
Starchy foods ^#^	0.80		0.96	
Fruit	0.44		0.94	
Vegetables	0.72		0.24	

* Statistically significant, at *p* < 0.05; ** *p* < 0.1 trend. ^#^ = containing gluten.

**Table 4 nutrients-16-00280-t004:** Vitamin intake values reported as median and interquartile range (IQR).

	CD-A V3	CD-R V3	CD-A V1	CD-R V1
Vitamin A **	404.82 (230.47–693.89)	559.6 (319.62–837.73)	436.39 (339.21–585.39)	497.32 (284.14–856.07)
Vitamin B1 *	0.96 (0.78–1.06)	0.91 (0.68–1.15)	0.95 (0.66–1.15)	0.95 (0.75–1.26)
Vitamin B2 *	1.16 (0.82–1.41)	1.12 (1.00–1.55)	1.1 (0.9–1.32)	1.11 (0.92–1.60)
Vitamin B3 *	15.57 (9.8–18.43)	15.06 (11.04–17.34)	13.94 (9.57–19.77)	14.47 (10.39–19.44)
Vitamin B5 *	1.31 (1.16–1.76)	1.45 (0.98–2.21)	1.65 (1.09–1.95)	1.44 (1.11–1.83)
Vitamin B6 *	1.34 (0.91–1.63)	1.38 (1.03–1.68)	1.15 (0.92–1.47)	1.38 (0.90–1.91)
Vitamin B8 *	5.98 (4.18–7.69)	6.9 (4.91–9.82)	7.48 (5.19–8.97)	5.74 (4.3–11.06)
Vitamin B9 **	151.64 (119.28–185)	155.79 (117.36–207.67)	142.68 (98.12–206.77)	167.7 (118.51–216.56)
Vitamin B12 *	1.6 (0.90–2.16)	1.51 (0.8–2.35)	1.51 (0.58–2.02)	1.41 (0.91–2.33)
Vitamin C *	48.93 (20.5–88.43)	50.47 (41.51–79.52)	45.36 (34.2–76.05)	57.38 (31.78–85.54)
Vitamin D *	2.49 (0.99–3.63)	2.37 (1.36–4.33)	1.76 (1.09–2.79)	1.90 (0.73–2.99)
Vitamin E *^#^	0.51 (0.26–0.66)	0.61 (0.26–0.76)	0.46 (0.32–0.75)	0.45 (0.30–0.79)
Vitamin k *	13.19 (8.95–17.6)	18.42 (9.76–51.13)	10.88 (6.43–25.88)	17.3 (8.54–36.09)
b-carotene **	726.36 (201.2–1493.8)	719.16 (338.79–1452.84)	716.04 (320.40–1338.57)	823.90 (314.54–1232.04)

UoM: * mg/day; ** μg/day. ^#^ as α-tocopherol.

**Table 5 nutrients-16-00280-t005:** Statistical significance of differences in vitamin intake in patients in active phase (CD-A) or in remission (CD-R) at V3 and V1. *p* values (two-tailed) are reported.

	V3 CD-A vs. CD-R	V1 CD-A vs. CD-R
Vitamin A	0.19	0.53
Vitamin B1	0.90	0.93
Vitamin B2	0.60	0.72
Vitamin B3	0.96	0.89
Vitamin B5	0.92	0.42
Vitamin B6	0.63	0.39
Vitamin B8	0.45	0.49
Vitamin B9	0.56	0.45
Vitamin B12	0.92	0.43
Vitamin C	0.56	0.55
Vitamin D	0.59	0.62
Vitamin E	0.53	0.96
Vitamin k	0.23	0.35
b-carotene	0.71	0.82

**Table 6 nutrients-16-00280-t006:** Mineral intake values reported as median and interquartile range (IQR).

	CD-A V3	CD-R V3	CD-A V1	CD-R V1
Calcium *	345.47 (290.36–510.18)	555.04 (434.38–690.43)	391.76 (290.67–526.19)	455.92 (302.24–669.57)
Sodium *	2500.25 (1950.9–2763.57)	1889.36 (1453.21–2882.46)	2240.23 (1385.68–3154.97)	2037.94 (1454.79–3250.07)
Potassium *	2171.81 (1643.54–2385.75)	2022.13 (1614.41–2799.31)	1964.24 (1514.08–2345.31)	2044.52 (1469.3–2676.46)
Phosphorus *	971.19 (811.83–1049.06)	1121.89 (839.01–1397.77)	1028.58 (764.02–1104.56)	1035.85 (837.84–1250.53)
Iron *	8.05 (5.84–10.56)	9.39 (7.55–12.06)	7.94 (6.11–10.49)	9.03 (7.53–10.82)
Zinc *	8.11 (6.26–9.72)	8.89 (6.82–10.13)	7.76 (5.25–9.49)	9.15 (5.8–11.58)
Magnesium *	112.14 (87.22–157.5)	137.00 (110.66–164.56)	127.37 (99.22–151.12)	129.14 (93.18–161.37)
Copper *	0.75 (0.44–0.9)	0.77 (0.59–1.01)	0.73 (0.44–0.86)	0.78 (0.57–0.95)
Selenium **	19.15 (13.27–24.85)	21.47 (14.89–32.28)	17.23 (14.23–23.47)	18.93 (11.88–25.33)
Chromium **	0 (0.00–0.0028)	0.0014 (0.00–0.0028)	0.00 (0.00–0.00143)	0.00 (0.00–0.00143)
Fluorine *	87.07 (60.27–153.55)	88.71 (44.43–171.69)	110.27 (50.78–169.97)	93.02 (30.85–203.84)
Iodine **	48.61 (37.94–81.21)	51.46 (35.03–66.94)	45.43 (33.94–59.38)	53.46 (29.34–73.85)
Manganese *	3.89 (1.52–8.40)	2.80 (1.38–8.93)	1.70 (1.13–3.67)	1.62 (1.06–4.94)
Molybdenum **	4.00 (2.37–9.22)	5.00 (2.37–9.03)	3.59 (1.93–7.91)	3.72 (2.01–6.89)

UoM: * mg/day; ** μg/day.

**Table 7 nutrients-16-00280-t007:** Statistical significance of differences in mineral intake in patients in active phase (CD-A), in remission (CD-R) at V3 and V1. *p* values (two-tailed) are reported.

	V3 CD-A vs. CD-R		V1 CD-A vs. CD-R	
Calcium	<0.01 *	CD-A < CD-R	0.27	
Sodium	0.32		0.73	
Potassium	0.66		0.38	
Phosphorus	0.14		0.38	
Iron	0.39		0.37	
Zinc	0.33		0.09 **	CD-A < CD-R
Magnesium	0.33		0.85	
Copper	0.34		0.19	
Selenium	0.43		0.85	
Chromium	0.3		0.88	
Fluorine	0.76		0.55	
Iodine	0.73		0.43	
Manganese	0.54		0.99	
Molybdenum	0.56		0.97	

* Statistically significant at *p* < 0.05; ** *p* < 0.1 trend.

**Table 8 nutrients-16-00280-t008:** Lipids intake values (mg/day) reported as median and interquartile range (IQR).

	CD-A V3	CD-R V3	CD-A V1	CD-R V1
Cholesterol	194.63 (165.84–285.71)	238.24 (166.39–321.36)	232.28 (143.78–301.25)	263.38 (169.3–320.02)
Saturated	20.52 (15.15–22.89)	24.65 (14.56–29.11)	20.42 (15.01–26.07)	22.41 (15.51–31.64)
Unsaturated	29.16 (24.91–37.87)	36.87 (26.42–44.45)	31.10 (24.21–37.03)	33.58 (23.29–48.73)
Polyunsaturated	7.26 (5.35–11.18)	9.32 (5.51–11.02)	7.50 (6.02–8.53)	7.69 (5.75–11.45)
Monounsaturated	22.37 (19.56–28.77)	26.36 (19.81–33.00)	24.05 (17.89–27.70)	25.58 (18.05–37.77)
C10:0 capric acid	0.11 (0.07–0.35)	0.34 (0.16–0.50)	0.19 (0.04–0.46)	0.25 (0.07–0.39)
C12:0 lauric acid	0.09 (0.03–0.15)	0.17 (0.09–0.23)	0.07 (0.02–0.22)	0.12 (0.04–0.19)
C14:0 myristic acid	0.33 (0.25–0.65)	0.67 (0.36–0.80)	0.38 (0.22–0.85)	0.50 (0.27–0.70)
C16:0 palmitic acid	4.05 (3.12–4.83)	4.17 (3.25–5.73)	4.09 (2.66–5.93)	3.56 (2.37–4.82)
C18:0 stearic acid	1.72 (1.28–2.09)	1.65 (1.37–2.89)	1.85 (1.09–2.79)	1.58 (0.96–2.21)
C20:0 arachidic acid	0.07 (0.03–0.13)	0.07 (0.04–0.11)	0.0 (0.03–0.10)	0.08 (0.04–0.14)
C14:1 myristoleic acid	0.01 (0–0.05)	0.01 (0–0.03)	0.02 (0.01–0.07)	0.01 (0.00–0.04)
C16:1 palmitoleic acid	0.42 (0.31–0.52)	0.37 (0.3–0.59)	0.46 (0.28–0.60)	0.36 (0.22–0.55)
C18:1 oleic acid	12.46 (10.05–16.00)	14.5 (9.5–17.78)	13.26 (8.15–15.67)	12.76 (9.02–18.86)
C20:1 eicosanoic acid	0.14 (0.08–0.18)	0.12 (0.07–0.22)	0.12 (0.09–0.18)	0.13 (0.07–0.22)
C22:1 erucic acid	0.01 (0.00–0.06)	0.01 (0.00–0.03)	0.00 (0.00- 0.02)	0.01 (0.00–0.03)
C18:2 linoleic acid	3.36 (2.49–4.47)	3.75 (2.68–4.47)	3.59 (2.60–4.87)	3.52 (2.28–5.36)
C18:3 α-linolenic acid	0.37 (0.24–0.49)	0.37 (0.24–0.51)	0.34 (0.27–0.52)	0.40 (0.23–0.6)
C20:4 arachidonic acid	0.12 (0.08–0.2)	0.12 (0.07–0.19)	0.14 (0.08–0.24)	0.12 (0.07–0.2)
C20:5 EPA	0.03 (0.00–0.04)	0.02 (0.01–0.04)	0.01 (0.00–0.03)	0.01 (0.00–0.03)
C22:6 DHA	0.01 (0.00–0.06)	0.02 (0.00–0.05)	0.00 (0.00–0.02)	0.01 (0.00–0.04)
Total ω3	0.42 (0.3–0.6)	0.45 (0.26–0.65)	0.42 (0.28–0.66)	0.42 (0.26–0.62)
Total ω6	3.59 (2.62–4.59)	3.87 (2.72–4.62)	3.79 (2.69–5.11)	3.67 (2.43–5.63)
ω3/ω6	0.12	0.12	0.11	0.11

**Table 9 nutrients-16-00280-t009:** Statistical significance of differences in lipid intake between patients in active phase (CD-A) and those in remission (CD-R) at V1 and V3. *p* values (two-tailed) are reported.

	V3 CD-A vs. CD-R		V1 CD-A vs. CD-R	
Cholesterol	0.33		0.49	
Saturated	0.38		0.44	
Unsaturated	0.55		0.48	
Polyunsaturated	0.37		0.57	
Monounsaturated	0.12		0.55	
C10:0 capric acid	0.04 *	CD-A < CD-R	0.85	
C12:0 lauric acid	0.05 **	CD-A < CD-R	0.62	
C14:0 myristic acid	0.28		0.59	
C16:0 palmitic acid	0.22		0.59	
C18:0 stearic acid	0.69		0.74	
C20:0 arachidic acid	0.4		0.33	
C14:1 myristoleic acid	0.77		0.06 **	CD-A > CD-R
C16:1 palmitoleic acid	0.78		0.43	
C18:1 oleic acid	0.63		0.87	
C20:1 eicosanoic acid	0.86		0.9	
C22:1 erucic acid	0.68		0.77	
C18:2 linoleic acid	0.86		0.94	
C18:3 α-linolenic acid	0.84		0.76	
C20:4 arachidonic acid	0.83		0.55	
C20:5 EPA	0.93		0.87	
C22:6 DHA	0.58		0.92	
Total ω3	0.83		0.90	
Total ω6	0.82		1.00	

* Statistically significant at *p* < 0.05; ** *p* < 0.1 trend.

**Table 10 nutrients-16-00280-t010:** Nutrient blood values reported as median and interquartile range (IQR).

	CD-A V3	CD-R V3	CD-A V1	CD-R V1
Cholesterol ^#^	169.0 (153.0–188.0)	181.0 (160.0–224.0)	155.0 (141.0–173.0)	162.0 (138.0–198.0)
Vitamin B9 ^§^	5.2 (4.7–7.5)	7.3 (5.8–9.6)	5.5 (3.0–6.9)	5.3 (3.8–6.9)
Vitamin B12 **	153.0 (120.0–279.0)	268.0 (202.0–333.0)	170.0 (154.0- 309.0)	243.0 (186.0–332.0)
Potassium *	4.0 (3.8–4.2)	4.1 (3.9–4.2)	4.1 (3.7–4.5)	4.0 (3.9–4.3)
Magnesium *	2.1 (2.0–2.2)	2.2 (2.1–2.4)	2.0 (1.8–2.1)	2.0 (1.9–2.1)
Iron ^●^	65.0 (31.0–115.0)	122.0 (97.0–167.0)	34.0 (20.0–47.0)	55.0 (36.0–90.0)
Sodium *	139.0 (138.0–141.0)	140.0 (137.0–141.0)	138.0 (138.0–140.0)	139.0 (138.0–141.0)
Phosphorus *	3.3 (2.9–3.7)	3.3 (3.1–3.6)	3.2 (2.7–3.6)	3.2 (3.0–3.5)
TAU *	83.5 (64.0–100.5)	97.0 (76.0–107.0)	64.0 (58.0–93.0)	87.0 (73.0–101.0)
ASP *	3.0 (2.0–4.0)	3.0 (2.0–4.0)	3.0 (3.0-4.0)	3.0 (3.0–5.0)
THR *	98.0 (91.5–124.5)	122.0 (111.0–136.0)	96.0 (87.0–126.0)	109.0 (97.0–142.0)
SER *	110.0 (102.5–122.5)	111.0 (98.0–127.0)	102.0 (90.0–111.0)	109.0 (84.0–117.0)
GLU *	671.5 (615.5–779.0)	806.0 (708.0–885.0)	639.0 (556.0–731.0)	651.0 (587.0–773.0)
PRO *	192.0 (135–224.0)	168.0 (150.0–198.0)	182.0 (145.0–246.0)	191.0 (164.0–247.0)
GLY *	219.0 (199.0–243.0)	235.0 (207.0–262.0)	226.0 (193.0–253.0)	234.0 (191.0–285.0)
ALA *	318.5 (287–381.5)	350.0 (293.0–396.0)	297.0 (238.0–367.0)	306.0 (263.0–376.0)
CIT *	31.5 (23.5–41.5)	39.0 (35.0–44.0)	30.0 (23.0–32.0)	33.0 (28.0–40.0)
VAL *	202.0 (156.0–241.0)	230.0 (198.0–256.0)	198.0 (171.0–217.0)	201.0 (171.0–222.0)
CYS *	40.5 (34.0–48.0)	47.0 (41.0–58.0)	38.0 (31.0–44.0)	40.0 (35.0–48.0)
MET *	21.5 (19.0–26.0)	27.0 (21.0–29.0)	21.0 (19.0–26.0)	21.0 (16.0–24.0)
ILE *	64.5 (54.5–74.0)	70.0 (58.0–76.0)	63.0 (52.0–73.0)	60.0 (50.0–68.0)
LEU *	135.5 (105.5–175.5)	159.0 (140.0–172.0)	134.0 (98.0–163.0)	131.0 (114.0–145.0)
TYR *	55.0 (52.5–64.5)	62.0 (55.0–75.0)	61.0 (48.0–74.0)	59.0 (50.0–67.0)
PHE *	55.5 (50–58)	65.0 (58.0–69.0)	59.0 (51.0–64.0)	59.0 (48.0–69.0)
ORN *	84.5 (75–102.5)	102.0 (70.0–122.0)	82.0 (62.0–105.0)	93.0 (74.0–118.0)
LYS *	183.5 (137.5–213.5)	211.0 (173.0–243.0)	168.0 (140.0–187.0)	174.0 (140.0–200.0)
HIS *	75.0 (63.5–88.0)	85.0 (78.0–96.0)	69.0 (52.0–77.0)	73.0 (63.0–82.0)
ARG *	42.0 (31.0–57.0)	56.0 (45.0–72.0)	37.0 (32.0–51.0)	47.0 (33.0–65.0)

Taurine (TAU), Aspartate (ASP), Threonine (THR), Serine (SER), Glutamic acid (GLU), Proline (PRO), Glycine (GLY), Alanine (ALA), Citrulline (CIT), Valine (VAL), Cysteine (CYS), Methionine (MET), Isoleucine (ILE), Leucine (LEU), Tyrosine (TYR), Phenylalanine (PHE), Ornithine (ORN), Lysine (LYS), Histidine (HIS), Arginine (ARG). CD-A: patients in the active phase. CD-R: patients in remission: ^#^ mg/dl; ^§^ ng/mL; ** pg/mL ^●^ μg/mL, * mmol/L.

**Table 11 nutrients-16-00280-t011:** Statistical significance of differences in nutrient blood values between patients in active phase (A) and patients in remission (R) at V1 and V3 visits. *p* Values (two-tailed) are reported.

	V3 CD-A vs. CD-R		V1 CD-A vs. CD-R	
Cholesterol	0.15		0.65	
Vitamin B9	0.03 *	CD-A < CD-R	0.54	
Vitamin B12	0.05 **	CD-A < CD-R	0.05 **	CD-A < CD-R
Potassium	0.63		0.79	
Magnesium	0.04 *	CD-A < CD-R	0.68	
Iron	<0.01 *	CD-A < CD-R	<0.01 *	CD-A < CD-R
Sodium	0.46		0.63	
Phosphorus	0.95		0.97	
TAU	0.23		0.02 *	CD-A < CD-R
ASP	0.67		0.78	
THR	0.01 *	CD-A < CD-R	0.35	
SER	0.64		0.76	
GLU	0.02 *	CD-A < CD-R	0.67	
PRO	0.32		0.58	
GLY	0.21		0.53	
ALA	0.34		0.60	
CIT	0.03 *	CD-A < CD-R	0.93	
VAL	0.04 *	CD-A < CD-R	0.46	
CYS	0.02 *	CD-A < CD-R	0.52	
MET	<0.01 *	CD-A < CD-R	0.92	
ILE	0.42		0.68	
LEU	0.09 **	CD-A < CD-R	0.99	
TYR	0.07 **	CD-A < CD-R	0.77	
PHE	<0.01*	CD-A < CD-R	0.95	
ORN	0.32		0.47	
LYS	0.06 **	CD-A < CD-R	0.71	
HIS	<0.01 *	CD-A < CD-R	0.28	
ARG	0.04 *	CD-A < CD-R	0.32	

* Statistically significant at *p* < 0.05. ** trend at *p* < 0.10. Taurine (TAU), Aspartate (ASP), Threonine (THR), Serine (SER), Glutamic acid (GLU), Proline (PRO), Glycine (GLY), Alanine (ALA), Citrulline (CIT), Valine (VAL), Cysteine (CYS), Methionine (MET), Isoleucine (ILE), Leucine (LEU), Tyrosine (TYR), Phenylalanine (PHE), Ornithine (ORN), Lysine (LYS), Histidine (HIS), Arginine (ARG).

## Data Availability

The data presented in this study are available on request from the corresponding author. The data are not publicly available due to ethical restrictions (patient confidentiality).

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
