# Peer review of "Nutritional Biomarkers for the Prediction of Response to Anti-TNF-α Therapy in Crohn’s Disease: New Tools for New Approaches"

_nutrients, 2024, doi:10.3390/nu16020280_

Round 1

Reviewer 1 Report

Comments and Suggestions for Authors

This paper describes a prospective exploratory study of Chron's patients, looking for possible interactions between their diets and treatment outcomes. It is designed and executed well.  There were reasonable subject numbers and the groups matched without clinical distinctions.  There weren't many drop-outs and it was conducted without any big  burdens on the patients. The results are well-documented and statistically explained.  There were a lot of negative findings which was OK because they provide evidence against ideas that ultra processed foods and red meat interfere with treatment success (type-2 error not withstanding).

The few positive findings were about about low iron, taurine and zinc. In my opinion these could be spurious findings due to the large number of variables.  Maybe the authors should comment on that.

The positive findings focused on iron, taurine and zinc as biomarkers.  The authors did well to place these findings in line with similar findings in earlier studies.

 I recommend the abstract be rewritten as follows to make it more clear.

After 3 6 months of anti-TNF-α, 4 patients dropped out of the study, leaving 29 patients were in clinical remission, while and 21 remained with active disease who remained until 6 months the same. The aim of this study was to identify nutrients whose intake or blood values may be associated with patients' response to biological therapy. In the diet, patients remaining with active CD showed very similar nutrients dietary intake compared to patients achieving remission except for a trend for lower starting zinc intake, below the reference value. In the blood, Instead, patients who did not respond to biological therapy showed significant lower plasma values of iron and taurine before starting biological anti-TNF- α treatment.

I assume the anti-TNFalpha Abs were given by injection, correct?  Assuming that more than one injection occurred over 6 months, and this was done at their clinic, how do I know there was no problem with treatment compliance?  Something about assuring compliance should be stated.  [It is helpful that the meds were not given by ingestion because that would've led to questions if the meds in the responders were simply up-taken better than in the non-responders.

On line 432 there are two repeated words:  In in...  

I'm not sure how this fits in, because low taurine in the blood may not relate to low zinc in the food, but taurine is known to be a zinc-binding ligand.

Author Response

Dear Colleague,

Thank you for your revision.

We have followed all your helpful suggestions.

Please see the following review followed point by point.

Comments and Suggestions for Authors

This paper describes a prospective exploratory study of Chron's patients, looking for possible interactions between their diets and treatment outcomes. It is designed and executed well.  There were reasonable subject numbers and the groups matched without clinical distinctions.  There weren't many drop-outs and it was conducted without any big  burdens on the patients. The results are well-documented and statistically explained.  There were a lot of negative findings which was OK because they provide evidence against ideas that ultra processed foods and red meat interfere with treatment success (type-2 error not withstanding).

The few positive findings were about low iron, taurine and zinc. In my opinion these could be spurious findings due to the large number of variables.  Maybe the authors should comment on that.

R: We thank the reviewer for this observation. A comment about this issue was included in lines 427-428

The positive findings focused on iron, taurine and zinc as biomarkers.  The authors did well to place these findings in line with similar findings in earlier studies.

R: We thank the reviewer for this appreciation. 

I recommend the abstract be rewritten as follows to make it more clear.

After 3 6 months of anti-TNF-α, 4 patients dropped out of the study, leaving 29 patients were in clinical remission, while and 21 remained with active disease who remained until 6 months the same. The aim of this study was to identify nutrients whose intake or blood values may be associated with patients' response to biological therapy. In the diet, patients remaining with active CD showed very similar nutrients dietary intake compared to patients achieving remission except for a trend for lower starting zinc intake, below the reference value. In the blood, Instead, patients who did not respond to biological therapy showed significant lower plasma values of iron and taurine before starting biological anti-TNF- α treatment.

R: Abstract was amended in line with Reviewer’s suggestions.

I assume the anti-TNFalpha Abs were given by injection, correct?  Assuming that more than one injection occurred over 6 months, and this was done at their clinic, how do I know there was no problem with treatment compliance?  Something about assuring compliance should be stated.  [It is helpful that the meds were not given by ingestion because that would've led to questions if the meds in the responders were simply up-taken better than in the non-responders.

R: We thank the reviewer for this comment. Therapy was administered by infusion. We added a statement about treatment and compliance in lines 96-98.

On line 432 there are two repeated words:  In in...  

R: The typo was corrected

I'm not sure how this fits in, because low taurine in the blood may not relate to low zinc in the food, but taurine is known to be a zinc-binding ligand.

R: We thank the reviewer for this interesting remark.

Reviewer 2 Report

Comments and Suggestions for Authors

The article presented to me for review, "Nutritional biomarkers for the prediction of response to anti- TNF-α therapy in Crohn's disease: new tools for new approaches" concerns the medically important topic of inflammatory bowel disease (Crohn's disease).

In the introduction, the authors presented the current state of knowledge on the topic presented in the publication.

I have no objections to the prepared plan of the clinical trial. 

The full protocol with all study procedures and analysis is available at the site www.clinicaltrial.gov, ClinicalTrials.gov Identifier: NCT02580864. This is a big advantage of the publication

They presented their results in 6 tables and 3 figures. 

The paper cited 26 items of current world literature.

The weakness of the paper is the small number of patients given to the study (54 cases) as well as the small control group (30 patients). In the current version of the paper, the authors are unable to change the number of cases studied. 

In addition, I believe that the current literature on the subject of the work should be presented to a greater extent. 

In my opinion,  the cited literature should be corrected.

Author Response

Dear Colleague,

Thank you for your revision.

We have followed all your helpful suggestions.

Please see the following review followed point by point.

 Comments and Suggestions for Authors

The article presented to me for review, "Nutritional biomarkers for the prediction of response to anti- TNF-α therapy in Crohn's disease: new tools for new approaches" concerns the medically important topic of inflammatory bowel disease (Crohn's disease).

In the introduction, the authors presented the current state of knowledge on the topic presented in the publication.

I have no objections to the prepared plan of the clinical trial. 

The full protocol with all study procedures and analysis is available at the site www.clinicaltrial.gov, ClinicalTrials.gov Identifier: NCT02580864. This is a big advantage of the publication

They presented their results in 6 tables and 3 figures. 

The paper cited 26 items of current world literature.

The weakness of the paper is the small number of patients given to the study (54 cases) as well as the small control group (30 patients). In the current version of the paper, the authors are unable to change the number of cases studied. 

R: We thank the reviewer for the comments. We are aware of the small size of the sample, but in order to have a starting population as homogeneous as possible we used rather restrictive inclusion and exclusion criteria. On the other hand, the limited number of patients allowed us to monitor them carefully during the study.

In addition, I believe that the current literature on the subject of the work should be presented to a greater extent. 

In my opinion,  the cited literature should be corrected.

R: We thank the reviewer for the advice. References were amended as suggested. Therefore, we added some references to our manuscript.